

# A novel no-sensors 3D model reconstruction from monocular video frames for a dynamic environment

Ghada M. Fathy[1,2], Hanan A. Hassan[1], Walaa Sheta[1], Fatma A. Omara[2,3] and Emad Nabil[2,4]

[1] Informatics Research Institute, City for Scientific Research and Technological Applications, SRTA-City, Alexandria, Egypt

[2] Department of Computer Science, Faculty of Computers and Artificial Intelligence, Cairo University, Giza, Egypt

[3] Faculty of Engineering, Heliopolis University, Cairo, Egypt

[4] Computer Science Department, Faculty of Computer and Information Systems, Islamic University of Madinah, Madinah, Saudi Arabia

Corresponding author
Ghada M. Fathy,
gfathy@srtacity.sci.eg,
eng.ghadafathy@gmail.com

## ABSTRACT

Occlusion awareness is one of the most challenging problems in several fields such as multimedia, remote sensing, computer vision, and computer graphics. Realistic interaction applications are suffering from dealing with occlusion and collision problems in a dynamic environment. Creating dense 3D reconstruction methods is the best solution to solve this issue. However, these methods have poor performance in practical applications due to the absence of accurate depth, camera pose, and object motion.This paper proposes a new framework that builds a full 3D model reconstruction that overcomes the occlusion problem in a complex dynamic scene without using sensors' data. Popular devices such as a monocular camera are used to generate a suitable model for video streaming applications. The main objective is to create a smooth and accurate 3D point-cloud for a dynamic environment using cumulative information of a sequence of RGB video frames. The framework is composed of two main phases. The first uses an unsupervised learning technique to predict scene depth, camera pose, and objects' motion from RGB monocular videos. The second generates a frame-wise point cloud fusion to reconstruct a 3D model based on a video frame sequence. Several evaluation metrics are measured: Localization error, RMSE, and fitness between ground truth (KITTI's sparse LiDAR points) and predicted point-cloud. Moreover, we compared the framework with different widely used state-of-the-art evaluation methods such as MRE and Chamfer Distance. Experimental results showed that the proposed framework surpassed the other methods and proved to be a powerful candidate in 3D model reconstruction.

## INTRODUCTION

Constructing a full 3D model from a complex dynamic scene data has many applications in motion capture, robot navigation, augmented reality, and autonomous driving. Moreover,

it aims to provide solutions to solve realistic interaction problems such as occlusion and collision. There are many challenges to reconstructing 3D models from dynamic scenes, such as predict accurate depth from sensors or a sequence of 2D RGB. To achieve that it needs to consider the camera pose, and the motion of dynamic objects during navigation.

There are many techniques in computer vision that introduce different image-based 3D modeling techniques such as simultaneous location and mapping (SLAM) (*Mur-Artal, 2017*), Multiview stereo (MVS) (*Kuhn, 2019*), photo tourism (*Furukawa, 2009*), and an RGB-D video-based method (*Keller, 2013*). These methods use the point cloud representation to represent a real-world scene. A point cloud representation can be used for 3D inspection as it renders detailed 3D environments accurately. The depth camera such as (RGB-D) cameras, e.g., Microsoft Kinect, is widely used to reconstruct 3D indoor scenes (*Chen, 2015*). However, Kinect-like scanning devices fail to capture reliable depth images from outdoor scenes. Moreover, RGB-D cameras may not be readily available in most robotics and mobile devices, and it may also introduce sensors noise.

A variant of solutions has been developed using different kinds of cameras, for example, monocular (*Tateno, 2017*; *Wang, 2018*), and stereo (*Hassan, 2017*). Monocular cameras are most widely used because they are cheap, least restriction, and most ubiquitous for robots and mobile devices. However, the existing monocular 3D model reconstruction methods have poor performance due to the lack of accurate scene depth and camera pose.

Therefore, most reconstruction directions resort to predict depth and camera pose using learning techniques. Generating a 3D point cloud from learnt depth and learnt camera pose might be the right choice that solves the problem of using expensive sensors and gives accurate results in a dynamic scene. Recently, deep neural network has been used in learning and succeeded to predict depth from a single image (*Liu, 2015*; *Laina, 2016*; *Casser, 2019*). One advantage of deep learning approaches is that the full scale can be predicted from a single image without the need of scene-based assumptions or geometric constraints.

Nowadays, several realistic interaction applications still have limitations to deal with the occlusion problem in a real dynamic environment without using expensive sensors. one of the most effective solutions is to generate dense 3D reconstruction for the whole scene. However, the interaction in a dynamic environment requires a true depth map and explicit not only to detect the camera localization but also consider the moving objects into the scene with each other and with static objects in the background.

The main objective of our framework is to create a smooth and accurate 3D point-cloud for a dynamic environment using accumulative information from a sequence of RGB video frames. This method is used in solving several problems such as occlusion and collision. Due to the expensive cost of multi-sensors data, a monocular camera is used instead and compensated by unsupervised learning techniques to be suitable for video streaming applications. The framework consists of two stages. In the first stage, the online refinement process adapts new environment by integrating 3D motion of moving objects with depth and camera ego-motion. In the second stage, a full 3D model is reconstructed using frame-wise point cloud fusion. Figure 1 illustrates the proposed framework details. The rest of the paper is organized as follows: The 'State-of-the-art' section describes the most relevant related work. The 'Proposed framework' section presents an overview of

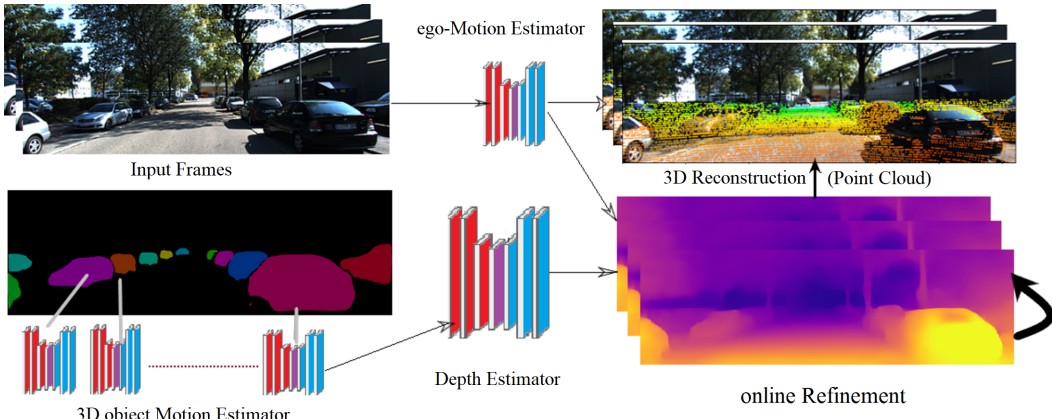

**Figure 1** **The proposed framework of 3D model reconstruction from monocular KITTI video images** (*Geiger, 2013*). KITTI dataset is under the Creative Commons Attribution-NonCommercial-ShareAlike 3.0 License. according to this link: http://www.cvlibs.net/datasets/kitti/.

the proposed approach. The 'Monocular 3D model reconstruction' section, explains the mathematical model for 3D model reconstruction. The 'Proposed 3D model reconstruction implementation' section illustrated dataset, experiment metrics. The 'Experimental Results' section prove the accuracy of the framework and give a comparison between our method and the state-of-the-art techniques. Finally, the conclusion and future work are explained in the last section.

# STATE OF THE ART

The development of the proposed framework has involved a review of research in the various computer vision field. This section is oriented towards three main subjects: (1) Estimate depth map from a single view. (2) Detect the camera position. (3) 3D reconstruction for a single object or multiple objects (full 3D scene) in a static and dynamic environment.

## Depth Estimation form single view

Scene depth estimation has gained increasing attention in the field of computer vision and robotics. Recently developed deep convolutional architectures for image-to-depth prediction has appeared fueled by the availability of rich feature representations, learned from raw data (*Eigen, 2015*; *Laina, 2016*; *Casser, 2019*). These approaches outperformed classical methods in terms of depth estimation accuracy (*Karsch, 2014*; *Liu, 2010*).

Numerous methods used supervised learning to estimate depth from a single view (*Wang, 2015*; *Ocal, 2020*). Despite supervised learning receives wide fame with depth prediction, it needs costly depth sensors for the training process. Therefore, many methods turn to using unsupervised learning image-to-depth techniques. Unsupervised depth prediction models have shown to be more accurate and get better performance than sensor-supervised methods (*Zhan, 2018*). Several consequent works result in good performance with the monocular setting (*Yang, 2017*; *Yin, 2018*). However, these methods are still incomplete

because they didn't handle object movements in dynamic scenes. These methods lead to failure as they cannot explain object motion in complex dynamic scenes.

## Camera pose estimation

Monocular Visual SLAM or Visual Odometry (VO) methods that include feature-based methods (*Mur-Artal, 2015*; *Klein, 2008*) and direct methods (*Engel, 2017*; *Forster, 2014*) are considered as a key tracking method for motion estimation. However, these methods lack of accurate depth estimation and are unable to handle pure rotation motions. CNN-SLAM (*Tateno, 2017*) and ORB-SLAM2 (*Mur-Artal, 2017*) solved monocular SLAM limitations by using deep neural networks to improve the scene depth. Nevertheless, these methods fail to give good performance in a dynamic scene. *Casser (2019)* have proposed a novel technique to solve the monocular depth and ego-motion problem by explicitly modeling 3D motions of moving objects, together with camera ego-motion, and adapts to new environments by learning with an online refinement of multiple frames.

## 3D reconstruction

The 3D reconstruction approaches are used for several domains. Table 1 summaries the main characteristics of the most relevant publications to our proposed framework

Nowadays, most state-of-the-art research used neural network techniques to reconstruct 3D objects from a single RGB image. Audrius et al. (*Kulikajevas, 2019*) applied hybrid neural network architecture to reconstruct polygonal meshes from a single depth frame using RGB-D sensors devices. Despite the RGB-D sensor still being capable to use. It is a lack to capture reliable depth images from the outdoor scene. Also, approach presented in *Li (2019)* used generation adversarial networks (GANs) to reconstruct a 3D object. GANs can generate 3D models by sampling from uniform noise distribution and get a significant performance. However, the authors succeeded to generate a 3D model using the GANs network but it is not tested to reconstruct full 3D scene (static or dynamic). Another approach focused on a single model in the medical domain is *Widya et al. (2019)*. They illustrated a 3D reconstruction technique for the whole stomach. Structure-from-Motion (SfM) with a monocular endoscope is used. The authors study the combined effect of chromo-endoscopy and color channel selection on SfM to increase the number of feature points and obtain more reliable reconstruction quality and completeness.

Scene reconstruction (static or dynamic) from video frames is one of the most important problems in computer vision field. This is because not only needs to deal with the camera pose, but also the object motions. Most of the RGB-D cameras have the natural advantage of reconstructing dense models, and many exciting fusion schemes are proposed in this domain (*Lee, 2016*; *Yan, 2017*). However, the received depth image from the depth camera contains noise or even outliers due to lighting conditions and spatially variant materials of the objects. *Wang (2018)* improved this limitation by combing learning depth from RGB-D datasets with monocular SLAM and frame-wise point cloud fusion to build a dense 3D model of the scene. They can reconstruct smooth and surface-clear on various examples with a dedicated point cloud fusion scheme.

The 3D reconstruction of a dynamic scene is more challenging than the static scene. This is because it does not only need to deal with the camera pose, but it also deals with the object

**Table 1 The main characteristics of the most relevant State-of-the-art.**

| Published | Single/Multiple frame | Single/Multiple object | Static/Dynamic object | Input type | Methods |
|---|---|---|---|---|---|
| *Kulikajevas (2019)* | Single frame | Single object | Static object | RGB-D sensor | Hyper neural network |
| *Kulikajevas (2019)* | Single frame | Single object | Static object | 3D Models | GANs neural network |
| *Widya et al. (2019)* | Multiple (2 image sequences) | Single object | Static object | Monocular endoscope | Structure from motion (SfM) |
| *Wang (2018)* | Single frame | Single object | Static object | RGB-D sensor | Monocular SLAM |
| *Yang (2020)* | Single frame | Multiple (full scene) | Static scene (remove dynamic objects) | Monocular RGB | Online incremental mesh generation |
| *Shimada (2020)* | Single frame | Single object | Dynamic object | Monocular RGB | Markless 3D human motion capture |
| *Peng (2020)* | Single frame | Single object | Dynamic object | Monocular RGB | GCN network |
| *Ku (2019)* | Single frame | Corp single object | Dynamic object | Monocular RGB | geometric priors, shape reconstruction, and depth prediction |
| *Lu (2020)* | Multiple (two consecutive point-cloud) | Multiple (full scene) | Dynamic objects | Outdoor LiDAR datasets | LSTM and GRU networks |
| *Weng et al. (2020)* | Single frame | Multiple (full scene) | Dynamic objects | Outdoor LiDAR datasets | Predict next scene using LSTM |
| *Akhter (2010)* | Single frame | Multiple objects | Dynamic objects | Monocular RGB | Structure from motion |
| *Fragkiadaki et al. (2014)* | Multiple frames | Single object | Dynamic object | Monocular RGB | Non-rigid structure-from-motion (NRSfM) |
| *Ranftl (2016)* | Multiple frames (two consecutive) | Multiple (full scene) | Dynamic object | Monocular RGB | Segments the optical flow field into a set of motion models |
| *Kumar, Dai & Li (2019)* | Multiple (2 frames) | Multiple (full scene) | Dynamic objects | Monocular RGB | Super pixel over segmentation |
| Proposed framework | Multiple (whole video frames sequence) | Multiple (full scene) | Dynamic objects | Monocular RGB | Unsupervised learning and point cloud fusion |

motion. In the past few years, there was a great interest to solve 3D scene reconstruction with moving objects using single or multiple Monocular camera RGB frames. Xingbin et al. (*Yang, 2020*), presented a real-time monocular 3D reconstruction system for mobile phone which used online incremental mesh generation for augmented reality application. For the 3D reconstruction process, they performed monocular depth estimation with a multi-view semi-global matching method followed by a depth refinement post-processing. Because the dynamic objects such as walking pedestrians or moving objects not support by multi-view geometry prerequisites, the authors deal with this problem by update the algorithm to remove the dynamic objects from the reconstruction mesh.

On the other hand, several applications focused on 3D reconstruction for a specific category of moving objects such as full or part of the human body. *Shimada (2020)* illustrated markless 3D human motion capture from monocular videos. They concentrated on challenging 3D motion observed, such as foot sliding, foot-floor penetration, and

unnatural body leaning. Also, *Peng (2020)* proposed 3D hand mesh reconstruction from a single RGB image. The authors relied on the graph convolution neural network (GCN) with two modules; hand localization and mask generation, to capture geometric details of 3D hand shape.

On the level of a dynamic outdoor domain, several domains are interested in predicting future 3d scenes from existing ones. *Lu (2020)* and Weng (2020) using neural networks such as LSTM and GRU to generate a full 3D point cloud from outdoor LiDAR datasets. The main idea is to use the motion-based neural network that integrates motion features between two consecutive point clouds.

Further, *Ku (2019)* introduced a monocular 3D object detection method that leverages proposals and shapes reconstruction. This method depends on three main processes; geometric priors, shape reconstruction, and depth prediction. The feature map is produced by image crop of the object and global context as input for the network. The orientation is predicted to estimate a proposal. Moreover, the point-cloud is generated and transformed into the global frame.

*Kumar, Dai & Li (2019)* illustrates a technique to build 3D reconstruction of a complex dynamic scene using two frames by applying super-pixel over-segmentation to the image. A generically dynamic (hence non-rigid) scene with a piecewise planar and rigid approximation are presented. Moreover, they reduced the reconstruction problem to a "3D jigsaw puzzle" which takes pieces from an unorganized "soup of super-pixels".

This work aims to create an accurate 3D scene model that recognizes every moving object from monocular RGB video frames without sensor data. The 3D reconstruction process learns information (depth map, camera pose, and object motion) from the current RGB frame, previous frame, and keyframes to detect changes during the object's motion. The 3D point cloud is continuously improved during this process by adding or removing some points according to a certain certainty filter. Studying moving objects from video frames solve several problems such as objects occlusion and collision in a dynamic outdoor environment such as augmented reality.

# THE PROPOSED FRAMEWORK

In this section, a novel framework for 3D dynamic scene reconstruction is proposed. This framework consists of two stages/modules. In the first stage, unsupervised learning is used to predict scene depth, camera pose, and object motion for a complex dynamic scene. Second, during the online refinement process, the previous information is used to create a point cloud for each single frame. Figure 2 explains the framework stages starting from video frames till the generation of the full scene point cloud.

## The online refinement process

This process was inspired by *Casser (2019)*. The advantage of this approach is not only predicting scene depth and camera pose, but also considering the motion of objects in a dynamic scene. Moreover, the framework gives better performance in outdoors than indoor scenes. The purpose of this approach is to represent the learnt geometric structure in the learning process by modeling the scene and the individual objects. From monocular

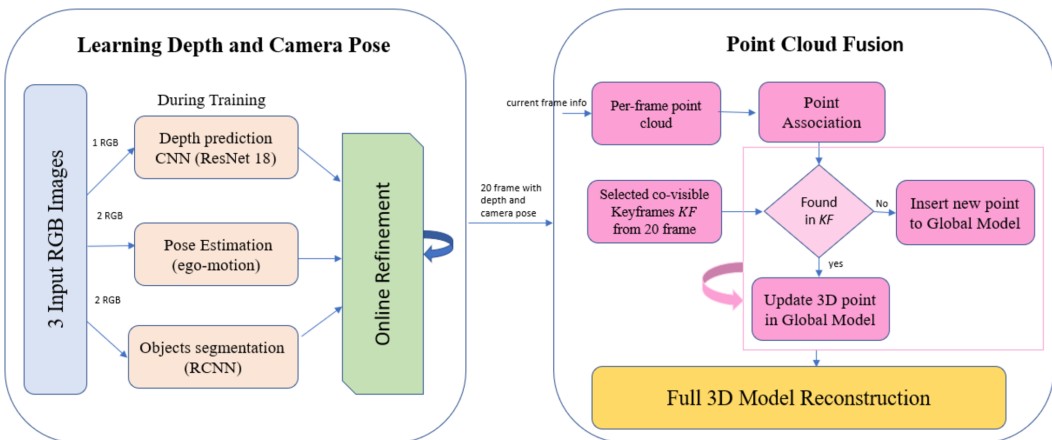

**Figure 2** The Proposed framework overview.

videos, the camera ego-motion and object motions are recognized. Furthermore, the online refinement method is used to adapt to learn on the fly to unknown domains. The depth function is a full convolution encoder–decoder architecture based on residual network (ResNet18) architecture (*He, 2016*). It begins with pre-trained weights on ImageNet (*Deng, 2009*), which produces a dense depth map from a single RGB frame. Moreover, the camera-motion neural network takes a sequence of two RGB images as input and generates an SE3 transform between the frames.

For object motion, the segmentation mask is used between two RGB images for every individual object. RCNN architecture with initialized pre-trained on the COCO dataset (*Lin, 2014*). In this stage, the framework predicts the transformation vector (Translation and rotation) for every single object in 3D space, which creates the detected object appearance in the respective target frame. The static background is generated by a single warp based on ego-motion.

## Point cloud fusion phase

The estimated RGB depth, camera pose, and object motion learnt in the previous phase is transformed to the frame-wise point cloud and later an accurate 3D model is reconstructed for the dynamic environment. The predicted 6-dimensional transformation vector (ego-motion) is used to convert the estimated depth into a unified coordinate space and then fused into an accumulated global model. The global model is a list of 3D points with correlating characteristics. For each frame, the per-frame point cloud is generated. The stability of each pixel in the current frame is checked with all co-visible keyframes. If corresponding points are found at any co-visible keyframe, the most stable point is merged with the new estimated point using a weighted average insertion. If not found, the estimated point is added to the global model as an unstable point. The global model is cleaned up overtime to remove outliers due to stability and temporal constraints.

The selection process of co-visible keyframes depends on the total number of video frames and positions of co-visible keyframes. Many experiments have been conducted

to select a suitable number of co-visible keyframes with their corresponding positions. As a result, it is found that five co-visible keyframes distributed over 20 frames gives an acceptable accuracy.

# PROPOSED MONOCULAR 3D MODEL RECONSTRUCTION

In this section, the proposed framework for 3D model reconstruction in a complex dynamic scene using monocular video frames will be illustrated. The proposed framework is divided into two main phases; unsupervised learning techniques phase for depth, camera pose and object motion, and point cloud frame-wise phase for a sequence of monocular video frames.

## Learning techniques phase for frame depth, camera pose, and object motion

The sequence of three RGB frames from monocular camera is used for the learning process $(I_1, I_2, I_3)$. The camera intrinsic matrix is defined as $K \in R^{3 \times 3}$. Depth prediction uses a single frame to produce a depth map. The depth map $D_i = \theta(I_i)$ is generated by a fully convolutional encoder–decoder architecture $\theta : R^{H \times W \times 3} \to R^{H \times W}$ (ResNet 18). In contrast, ego-motion network $\psi E : R^{2xHxwx3} \to R^6$ takes sequence of two frames and produces a SE3 transform vector (Translation and Rotation) between frames. Different warping operation in one frame is used to adjacent one in sequence. It allows predicting how the scene seems like with a different camera viewpoint. Using different frame warping operator $\varnothing(I_i, D_i, E_{i \to j}) \to \hat{I}_{i \to j}$., where $\hat{I}_{i \to j}$ is the reconstructed $j$th image. This approach able to change any source RGB-image $I_i$ into $I_j$ given corresponding depth estimate $D_j$ and an ego-motion estimate $E_{i \to j}$.

In practice, $\varnothing$ performs the warping by reading from transformed frame pixel coordinates. The projected coordinates are calculated by setting $\hat{I}_{i \to j}^{x,y} = \hat{I}_{i \to j}^{\hat{x}\hat{y}}$ where $[\hat{x}, \hat{y}, 1]^T = KE_{i \to j}(D_j^{x,y}.K^{-1}[x, y, 1]^T)$. The reconstruction loss for this approach is calculated as mentioned in *Casser, (2019)*

The object motion model $\psi M$ is used to predict the motion of individual objects in 3D space. Similar to ego-motion architecture, it used a sequence of two RGB frames. The object motion applied a segmentation mask (RCNN architecture) for individual objects into the dynamic scene. The transformation vector per object is learned, which creates the detected object appearance in the respective target form. According to the ego-motion model, the static background is generated and then all segmented objects are added by their appearance. The advantage of this approach is not only modeling objects in 3D space, but also learning their motion on the fly.

## 3D model reconstruction with point cloud fusion

After depth, camera pose and object motion are predicted in the previous stage. It is time to reconstruct a full 3D scene model using point cloud fusion. The point cloud generation is inspired by ORB-SLAM2 (*Mur-Artal, 2017*; *Wang, 2018*). *Wang (2018)* used point cloud fusion to generate 3D model reconstruction suitable for augmented reality applications. The advantage of this approach is that it is easy to apply and convenient for real-time

applications. The limitation of this approach is being limited to a static environment and is not tested in a dynamic environment. The proposed framework alleviates this limitation to consider objects motion' in dynamic environments. The per-frame point cloud is reconstructed by estimating depth $D_i$ for a single frame, the final Motion $E_m^F$ that is a combination of individual moving objects $\psi M_o$ and Camera motion $\psi E_{i \to j}$. Once the intrinsic camera calibration matrix $K$ is given, the per-frame point cloud is calculated as following:

$$p_i = (E_m^F)^{-1} \pi(u, D_i) \tag{1}$$

Where $u$ denote as homogeneous representation of a pixel $u = (x, y, 1)^T$ and $\pi(u)$ is the back projection from image to camera coordinate, $\pi(u, D_i) = K^{-1} D_i u$.

## Point association

The system holds a set of co-visible of keyframe $k_1, k_2 \ldots k_n \in KF$ selected according to the length of monocular video frames on the online refinement process. The visibility of pixels is checked by mapping each pixel of the current frame with all co-visible keyframes.

$$u^k = f\left(K(E_m^F)^k \pi\left(u^i\right)\right) \tag{2}$$

where $f(x) = (\frac{x}{z}, \frac{y}{z})^T$, also, maintain such a mapping from every keyframe pixel to its corresponding 3D point $M : p_i \to u^i \to u^k \to P$. where P is a global model.

To create a smooth 3D point cloud and filter out the noise from the generated points, the probabilistic filter is used. Each 3D point in global model P is represented by $p_i^n$, and the confidence counter $C_c$ is defined as how often the 3D point is observed in co-visible keyframes. The $C_c$ determines if 3D point evolves from unstable to stable state. Our weighted average is calculated by applying a Gaussian weight to the current depth measurement as $w_A = e^{-\gamma^2/\sigma^2}$ where $\gamma$ is the normalized radial distance of $D_i$ from the center of the camera, and $\sigma = 0.6$. The new observation available in the latest frame $i$ according to the following equations:

$$p_i^n = \left(w_A p_i + w^0 (E_m^F)^{-1} \pi\left(u^i\right)\right) / (w_A + w^0) \tag{3}$$

$$C_C^n = \left(w_A C_c + w^0 \|(E_m^F)^{-1} \pi\left(u^i\right) - p_i\|\right) / (w_A + w^0) \tag{4}$$

$$w_A^n = \min(w_A + w^0, W_\varepsilon) \tag{5}$$

Where $p_i^n$ means the newly updated point, $w^0$ is a constant equal to 1 and $W_\varepsilon$ is the truncation threshold equal to 100 (*Wang, 2018*).

Figure 3 represents the pseudo code of 3D model reconstruction process. The point association start from line 8 to 25; in which the stability of each pixel in the current frame is checked with all co-visible keyframes. If the corresponding points are found, the point is updated and set as stable according to the value of its corresponding confidence counter. If it is not found, the estimated point is added to the global model as an unstable point and inserted to point map as new seed. Probabilistic noise filter is applied at line 27 to maintain only stable points.

**Generate a 3D Reconstruction for dynamic scene**

**Input**: $\mathbb{P} \leftarrow$ global Model, hash map contains 3D point cloud, confidence counter, average weight, and point status (Stable, Unstable) with length s

$\quad$ $\mathbb{M} \leftarrow$ point Map, mapping of [x, y] and pointID for $u^k$ with length s

$\quad$ L$\leftarrow$ the $u^k$ dimension

$\quad$ F $\leftarrow$ total number of frames

$\quad$ KF $\leftarrow$ total number of Co-visible keyframe

```
1  for i = 1 → F do
2  |  % project current frame to world coordinates
3  |  p_i = (E_m^F)^{-1} π(u, D_i)
4  |    for k=1 → KF do
5  |    |  % project world coordinates to co-visible keyframe
6  |    |  p_i → u^i → u^k
7  |    |  u^k = f(K (E_m^F)^k π(u^i))
8  |    |    for j=1 → L do % point association
9  |    |    | if j in M:
10 |    |    | | % point is visible
11 |    |    | | % current frame-wise 3D point associated with the 3D global model
12 |    |    | | % update point info
13 |    |    | | p_i^n = (w_A p_i + w^0 (E_m^F)^{-1} π(u^i)) / (w_A + w^0)
14 |    |    | | C_C^n = (w_A C_c + w^0 ||(E_m^F)^{-1} π(u^i) - p_i||) / (w_A + w^0)
15 |    |    | | w_A^n = min (w_A + w^0, W_s)
16 |    |    | | if C_C^n < stable_threshold
17 |    |    | |   % point is stable
18 |    |    | | else
19 |    |    | |   % point unstable
20 |    |    | else
22 |    |    | | insert p_i to P % with all point info
23 |    |    | | insert n to M
24 |    |    | | |___
25 |    |    | |_____
26 |    |    |___
27 |    % remove unstable points from P
28 |___________________________________________
```

**Output**: accumulated global model $\mathbb{P}$ for 3D reconstruction

**Figure 3** The pseudo code of 3D model reconstruction process.

# PROPOSED MONOCULAR 3D MODEL RECONSTRUCTION IMPLEMENTATION

## Dataset and implementation details

The proposed framework was evaluated by using KITTI dataset (*Geiger, 2013*). The KITTI dataset is the most recent dataset used in different applications because it contains different objects, and it is considered a complex dynamic environment. The KITTI dataset has LIDAR sensor readings for evaluation only. It is used to evaluate predicted depth and ego-motion. Moreover, the KITTI 3D point cloud is used as ground truth to evaluate the proposed 3D model reconstruction model.

The number of points per scan is not constant, on average each frame has a size of ~1.9 MB which corresponds to ~120,000 3D points.

The proposed framework is divided into two modules/phases as mentioned in 'The Proposed Framework' . The first module is responsible for predicting depth and ego-motion using unsupervised learning in a dynamic scene. The code implemented using TensorFlow, the actual size of input images is $1,224 \times 368$, the images are resized to $416 \times 128$, the same setting which described in *Casser (2019)* is used such as learning rate 0.0002, L1 reconstruction weight 0.85, SSIM weight 0.15, smoothing weight 0.04, object motion constraint weight 0.0005, and batch size 4. The dataset is divided into training, validation, and testing (30542 monocular triplets for training, 3358 for validation, and 698 for testing). The framework has been executed on high-performance computing (HPC), PowerEdge Dell R740 (2x intel Xeon Gold 6248 2.5G) with Tesla V100 GPU. The second module generates a 3D point cloud to reconstruct a dynamic scene. This module is implemented in a framework of TensorFlow using Python, OpenGL, and open3D. During the online refinement process, 20 frames are selected to generate a 3D point cloud for each frame and finally integrated into one 3D model of the scene.

## Evaluation metrics

The evaluation process used three methods:

1. Localization Accuracy Error $L_E$, FPE (False positive error), and FNE (False Negative error) which were proposed by Refs (*Hafiz, 2015*).

**Localization Accuracy,** $L_E$ is defined as the amount of deviation of the detected point from a ground truth point position. Let ground truth data set denoted by $G_T$, ground truth point denoted by $p_g \in G_T$, $P_p$ is predicted points by the proposed technique. $N_G$ is the number of points in $G_T$, $N_p$ is the number of points in $P_p$, and $C_r(p_g)$ is geodesic distance over a region, which is centered by point $g$ and has radius of $r$. The data which is contained in $C_r(p_g)$ can be defined as:

$$C_r(p_g) = \{p_c \in P_p | Min(dis(p_g, p_c)) < r\}$$

where $dis(p_g, p_c)$ is the Euclidian distance between the two points $p_g$ and $p_c$, and $r$ is the Maximum correspondence points-pair distance which controls the localization error. $p_c$ is considered to be correctly detected if there exists a detected point $p_c \in P_p \bigcap C_r(p_g)$ such that $p_c$ is the minimum distance between the points $p_g$ and $p_c$. The $L_E$ defined as follows:

$$L_E = \sqrt{\frac{1}{N_C} \sum_j^{N_C} dis\left(p_{g_j}, p_{c_j}\right)}$$

where $N_C$ is the number of correctly detected points in $G_T$. The FNE at localization error tolerance $r$ is defined as:

$$FNE(r) = 1 - \frac{N_C}{N_G}$$

The FPE at localization error tolerance $r$ is defined as:

$$FPE(r) = \frac{N_F}{N_p}$$

The number of false positives is normalized with the number of all true negatives, where $N_F$ is the number of false positives, and yields to

$$N_F = N_p - N_C$$

2. **Registration** 3D point cloud between the output of the Velodyne laser scanner (ground truth) and the proposed technique which generates a 3D point cloud from the predicted depth and predicted ego-motion. Global registration (*Zhou, 2016*) and Iterative Closest Point ICP point-to-point (*Rusinkiewicz, 2001*; *Paul, 1992*) are used. Moreover, evaluate the registration by calculating Fitness function which is used to measure the overlapping area (the number of inlier correspondences/number of points in ground truth). The Higher value of fitness is better. While for the Root Mean Square Error RMSE of all correspondences in range of $r$, the lower is better. 3- **Accuracy** is reported using mean relative error (MRE). Which defined as

$$MRE = \frac{1}{P} \sum_{i=1}^{P} \frac{|z_{gt}^i - z_{est}^i|}{z_{gt}^i}$$

Let $z_{gt}^i, z_{est}^i$ are the ground-truth depth and the estimated depth respectively with $P$ as the total number of 3D point Cloud. MRE is state of the art metric used to compare the proposed framework with several monocular dynamic reconstruction methods. Another used metric is **Chamfer Distance (CD)** between the ground truth $P_{gt} \in R^{N \times 3}$ point cloud and the estimated point cloud $P_{est} \in R^{N \times 3}$. Chamfer Distance (*Lu, 2020*) is a regularly used metric to measure the similarity between two-point clouds, which is define as:

$$CD = \frac{1}{N} \sum_{x^{\wedge i} \in P_{est}} min_{x^j \in P_{gt}} \left| x^{\wedge i} - x^j \right| + \frac{1}{N} \sum_{x^j \in P_{gt}} min_{x^{\wedge i} \in P_{est}} \left| x^{\wedge i} - x^j \right|$$

## EXPERIMENTAL RESULTS

### Localization accuracy

In this section, the proposed framework has been evaluated using different techniques. At first, localization error $L_E$, FNE, and FPE using different localization error tolerance $r$ between ground truth and predicted 3D point cloud is calculated.

Figure 4 illustrates the average of localized error, FNE, and FPE with $r$ in range 0.04 to 0.16 for 20 frames, and the output of the online refinement process. Form Fig. 4, it is found that the proposed framework is succeeded in finding points close to ground-truth points with low localization error. Moreover, decreasing in FNE indicates that the framework catches the nearest points with a low localization error, while a rapid drop in FPE means that the framework does not return excessive interest points.

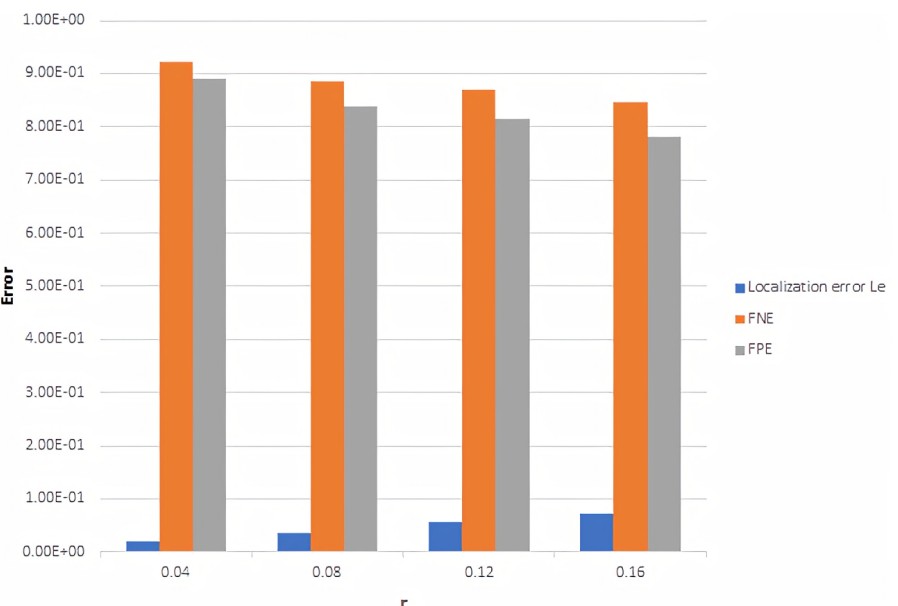

**Figure 4** Localization error, FNE and FPE with different *r* value.

## Point cloud registration

The second approach using 3D point cloud registration between ground-truth and predicted points is used to check the accuracy of 3D reconstruction. Figure 5 shows a selected frame from a sequence of 20 frames registered with ground-truth with two different points of view. The number of predicted 3D point cloud depends on the number of 2D pixels of the RGB frame. In our case, the input frame size is 416 × 128 (in range of 50,000 points), counter to the 3D point cloud of ground truth that collected from the Velodyne laser scanner is in the average of 120,000 points. As shown in Fig. 5, the range and density of the predicted 3D point cloud are less than the ground truth. Therefore, the predicted 3D point cloud is closed to the ground-truth in the selected area.

To evaluate the registration between ground-truth and predicted 3D point cloud, we used the state of the art algorithms such as Global registration (*Zhou, 2016*) and ICP point-to-point. (*Rusinkiewicz, 2001*; *Paul, 1992*). Figures 6 and 7 illustrate the average of RMSE and fitness of registration for 20 frames on online refinement process using Global registration and ICP point-to-point with different threshold.

As shown in Figs. 6 and 7, ICP point-to-point registration gets the lowest RMSE and higher fitness between ground truth and predicted 3D point cloud. This, is because the ICP point to point technique usually runs until convergence or reaches a maximum number of iterations (we used the default iteration 30). This indicates that the proposed framework succeeds to cover large number of an overlapping areas with a small mean square error.

Figure 8 gives more details about ICP point-to-point registration during 20 frames with an acceptable RMSE and stander deviation with the increasing of *r* value.

Figure 9 illustrates the 3D point cloud after mapping on RGB frames selected from different videos. Figure 9B is a referee to the ground-truth point, and Fig. 9C to the

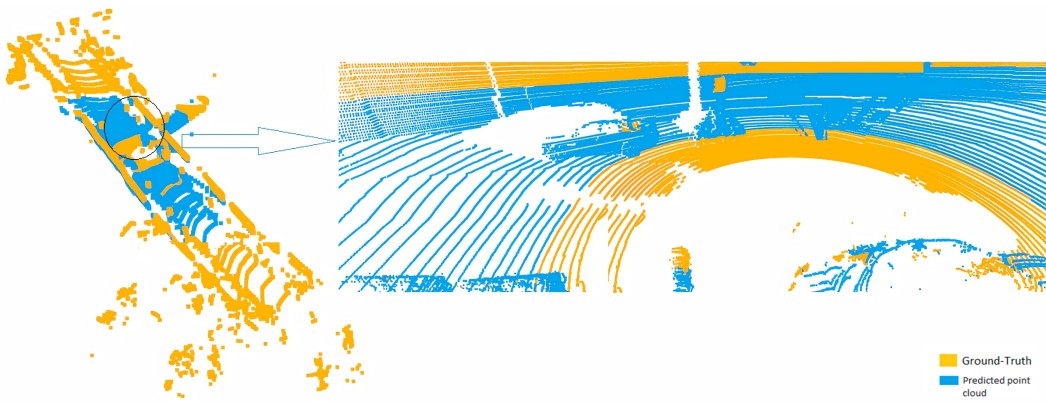

**Figure 5** Registration between ground-truth (yellow) and predicted 3D point cloud (blue).

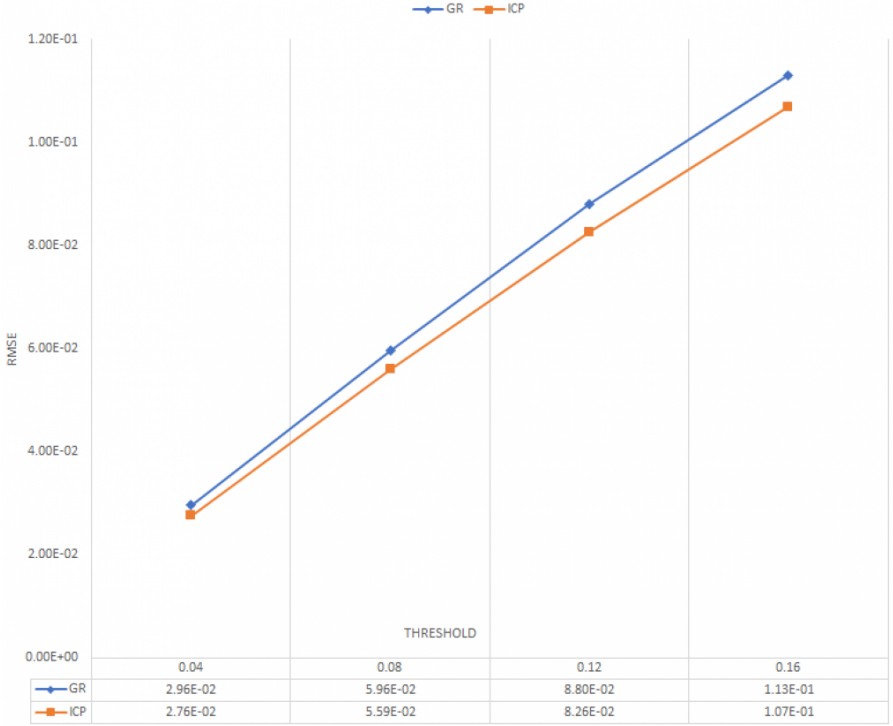

| | 0.04 | 0.08 | 0.12 | 0.16 |
|---|---|---|---|---|
| GR | 2.96E-02 | 5.96E-02 | 8.80E-02 | 1.13E-01 |
| ICP | 2.76E-02 | 5.59E-02 | 8.26E-02 | 1.07E-01 |

**Figure 6** Average RMSE for 20 frames.

predicted point from our framework. The performance of the proposed framework is compared with the state-of-the-art methods (*Kumar, 2017*), which reported that MRE on KITTI dataset and with several monocular dynamic reconstruction methods, such as the Block Matrix Method (BMM) (*Dai, 2014*), Point Trajectory Approach (PTA) (*Akhter, 2010*), and Low-rank Reconstruction (GBLR) (*Fragkiadaki et al., 2014*) , Depth Transfer

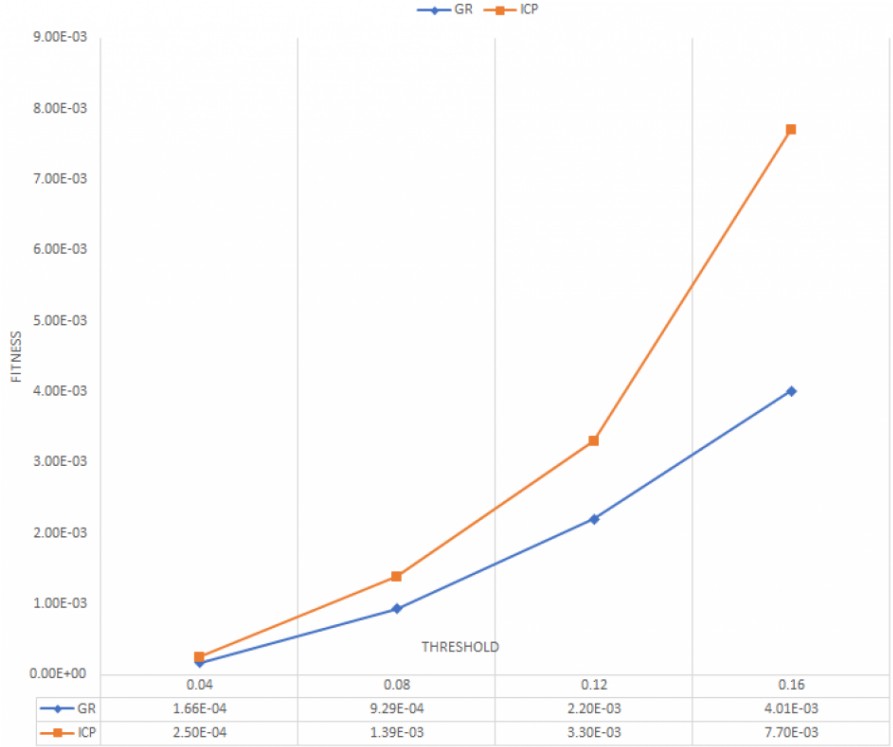

| | 0.04 | 0.08 | 0.12 | 0.16 |
|---|---|---|---|---|
| GR | 1.66E-04 | 9.29E-04 | 2.20E-03 | 4.01E-03 |
| ICP | 2.50E-04 | 1.39E-03 | 3.30E-03 | 7.70E-03 |

**Figure 7** Average fitness for 20 frames.

**Table 2** The improvement percentages in MRE error between proposed framework and the state-of-the-art.

| Approach | Improvement Percentage % |
|---|---|
| BMM | 75.02775 |
| PTA | 83.58463 |
| GBLR | 82.70561 |
| DT | 83.49633 |
| DMDE | 54.39189 |
| DJP | 46.76656 |

(DT) (*Karsch, 2012*), and (DMDE) (*Ranftl, 2016*). Note that we used the reported result in *Kumar (2017)* as its implementation is not available publicly.

Figure 10 shows that the proposed framework delivers consistently superior reconstruction accuracy on the KITTI dataset. Using unsupervised learning to predict scene depth and camera pose is a strong point of the proposed framework for generating an accurate 3D model reconstruction. Table 2 shows the improvement percentages between the proposed framework and the state-of-the-art methods.

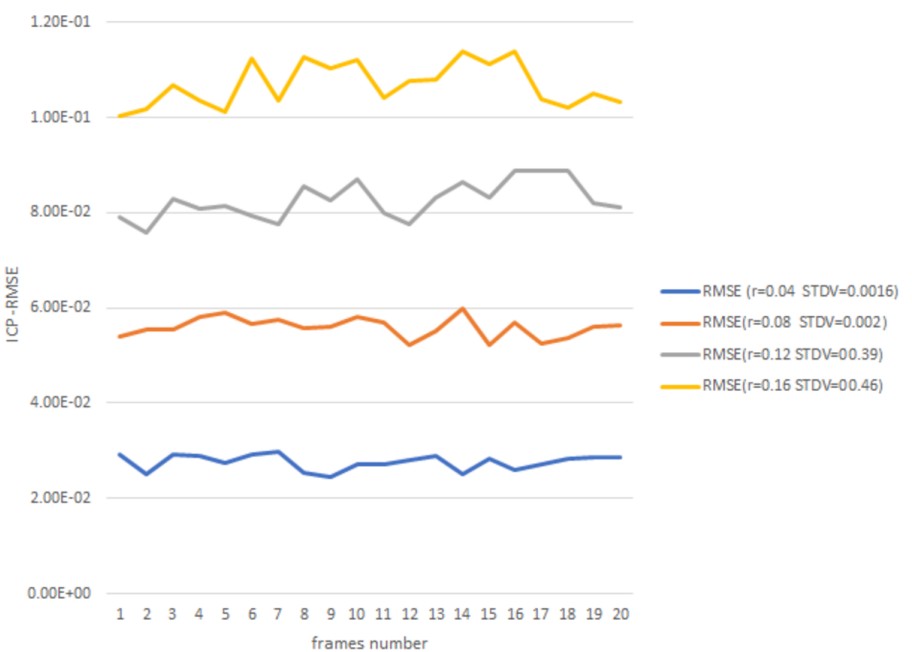

**Figure 8** ICP-RMSE for 20 frames.

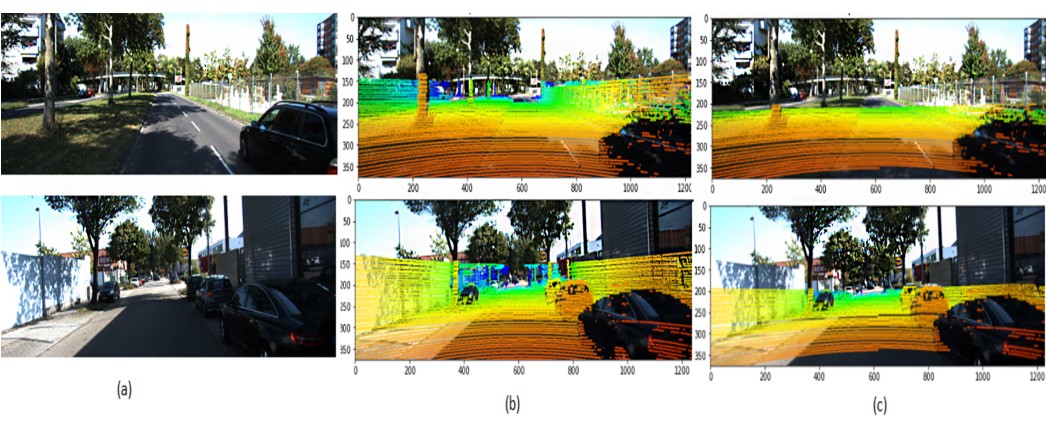

**Figure 9** 3D point cloud mapped to 2D KITTI image (*Geiger, 2013*). (A) Selected input frame; (B) ground truth; (C) predicted points. The KITTI dataset is under the Creative Commons Attribution-NonCommercial-ShareAlike 3.0 License. http://www.cvlibs.net/datasets/kitti/.

## Chamfer distance

We calculate the Chamfer Distance (CD) between the point cloud and ground truth on the KITTI dataset and compare it with state-of-the-art modules proposed in *Lu (2020)*. The main idea of this metric is to predict future frames given the past point cloud sequence based on a motion-based neural network named MoNet. Two neural networks are used to predict scene point-cloud LSTM and GRU.

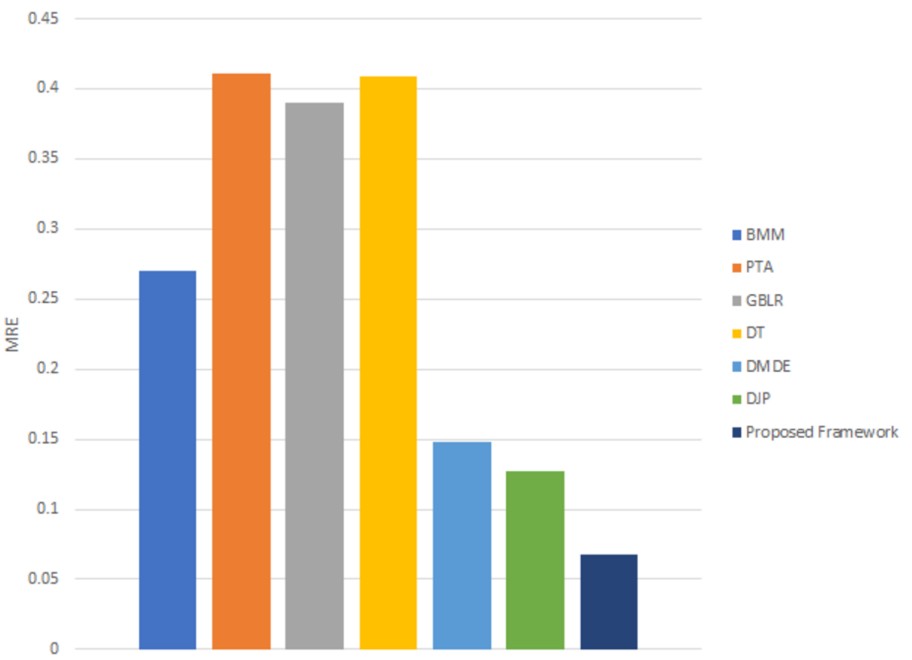

**Figure 10**  MRE for 3D reconstruction using different techniques on the KITTI dataset.

**Table 3**  Comparison using Chamfer Distance between the proposed framework and state-of-the-art.

**KITTI**

| Model | Chamfer distance |
| --- | --- |
| MoNet (LSTM) | 0.573 |
| MoNet (GRU) | 0.554 |
| *Proposed framework* | *0.491* |

The main idea of these methods using point-cloud as an input for neural networks to estimate future frames. The MoNet integrates motion features into the prediction pipeline and combines them with content features. In this metric, we used the average CD for 5 frames to match with the module (*Lu, 2020*). As shown in Table 3, the CD of our framework is slightly better than Mon (LSTM) and Mon (GRU) methods.

Finally, the experimental results show how the proposed framework achieves an accurate 3D reconstructed point-cloud model from monocular RGB video frames without using expensive sensors. Several evaluation metrics are measured, such as Localization error, RMSE, and Fitness between ground truth and predicted point-cloud. Finally, the experimental results show how the proposed framework achieves an accurate 3D reconstructed point-cloud model l from monocular RGB video frames without using expensive sensors. Several evaluation metrics are measured, such as Localization error, RMSE, and Fitness between ground truth and predicted point-cloud. Moreover, we achieved 46% improvement in MRE error compared with the state-of-the-art method

DJP. Besides, 11% and 14% improvement using chamfer distances metric compared with MonNet (GRU) and MoNet (LSTM) respectively.

### The limitations

The success of the presented framework depends on the accuracy of learning parameters such as depth map, camera pose, and object motion. In the case of the learning parameters are not processed accurately in phase one, the 3D reconstruction will fail. The other major limitation is the overall execution time. Because the generation of a 3D point cloud depends on accumulative matching between the current frame and a group of keyframes, this process takes up to 15 min. Moreover, the proposed framework had reconstructed a point cloud from a sequence of 20 RGB video frames which is considered a short sequence. However, this limitation could be overcome by using parallel programming to handle the most time-consuming part of the 3D point cloud reconstruction as mentioned in the future work section.

## CONCLUSION AND FUTURE WORK

This paper proposes a novel framework for 3D model reconstruction from monocular video frames for a dynamic environment. The framework didn't use any sensor data, which is costly and sometimes noisy. The results showed that the proposed framework is capable of generating smooth and accurate 3D point-cloud for a dynamic environment using cumulative information of a sequence of RGB video frames. Different evaluation metrics are used such as Localization error and RMSE with average values of 0.05 and 0.067 respectively between ground truth and predicted point-cloud. Moreover, the increase in fitness value indicates that the proposed framework succeeded to cover a large number of overlapping areas with a small mean square error.

Furthermore, a comparison between the proposed framework and state-of-the-art method using MRE compared with the DJP technique and Chamfer Distance compared with two MoNet techniques with an improvement of 46% ,11%, and 14% respectively. In the future, we will be concerned with improving the overall execution time to make it able to deal with real-time applications such as augmented reality by applying several optimization techniques using state-of-the-art GPU and CUDA platforms. In addition to, test 3D model reconstruction over long sequences of RGB frames.

### Funding

This research work was supported by the Egyptian Academy of Scientific Research and Technology (ASRT) JESOR Grant. This project was entitled 'Resource-effective Cloud Data Center: Developing a dynamic data center management tool'. The funders had no role in study design, data collection and analysis, decision to publish, or preparation of the manuscript.

### Grant Disclosures

The following grant information was disclosed by the authors:

Egyptian Academy of Scientific Research and Technology (ASRT) JESOR.

## Competing Interests
The authors declare there are no competing interests.

## Author Contributions
- Ghada Mohamed Fathy conceived and designed the experiments, performed the experiments, analyzed the data, performed the computation work, prepared figures and/or tables, authored or reviewed drafts of the paper, and approved the final draft.
- Hanan Ali Hassan and Emad Nabil conceived and designed the experiments, analyzed the data, authored or reviewed drafts of the paper, and approved the final draft.
- Walaa Sheta and Fatma Omara conceived and designed the experiments, authored or reviewed drafts of the paper, and approved the final draft.

## Data Availability
The raw code is available in the Supplemental Files.

We also used the KITTI dataset. It is available online: Geiger, A. P. (2013). Vision meets robotics: The kitti dataset. The International Journal of Robotics Research 32, no. 11, 1231-1237. http://www.cvlibs.net/datasets/kitti/raw_data.php.

## Supplemental Information
Supplemental information for this article can be found online at http://dx.doi.org/10.7717/peerj-cs.529#supplemental-information.

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
