# Peer review of "A novel no-sensors 3D model reconstruction from monocular video frames for a dynamic environment"

_PeerJ Computer Science, doi:10.7717/peerj-cs.529_

## Round 0.1 · original submission · Major Revisions

Revise the manuscript as suggested by the reviewers.

Reviewer 1 ·

Basic reporting

The authors need to clearly state their motivation, goal and contributions at least in the abstract and the introduction section of the paper.

Literature review must be improved as it lacks details and in-depth comparison to other state of the art techniques and applications,e.g.: https://www.mdpi.com/1424-8220/19/7/1553
https://doi.org/10.2991/ijcis.d.190617.001
https://doi.org/10.1109/IECON43393.2020.9255086
https://doi.org/10.1109/JTEHM.2019.2946802
https://jivp-eurasipjournals.springeropen.com/articles/10.1186/s13640-018-0253-2

Rewrite adding a section on each item, how it works, how your solutions differs, what has been achieved, end with how your goals correlate.

Novelty should be better explained as similar solutions do exist:
https://openaccess.thecvf.com/content_CVPR_2019/html/Ku_Monocular_3D_Object_Detection_Leveraging_Accurate_Proposals_and_Shape_Reconstruction_CVPR_2019_paper.html
https://doi.org/10.1109/TVCG.2020.3023634
https://dl.acm.org/doi/10.1145/3414685.3417877
https://link.springer.com/article/10.1007/s00371-020-01908-3

Figure 9 is very low res and hard to see.

Figure 3 should be replaced with activity diagram. Add all configuration parameters.

Get rid of trivial text on the methods, a link to original source is enough. Your reader will be familiar as the approach is not unheard.

Experimental design

The experiments should be contextualized better (the reader should not be left to assume that they will get their own conclusions). The experiments should be described more clearly (e.g. set up and carry out process, results in raw format, etc.). How was the accuracy of the model evaluated? By what metrics? Expert knowledge? Add full and rigorous statistical reliability analysis proving that recreated model is accurate. Add full performance metrics.

Validity of the findings

Article contains no direct comparison to other works with KITTI dataset (and there has been quite a few)

Code attached does not work.

Conclusions are quite generic. Focus on the results only. Move the rest to discussions.

Reviewer 2 ·

Basic reporting

Article proposes a two-tier depth estimation and pointcloud reconstruction capable of rebuilding occluded data from a complex scene. Where depth, camera pose and egomotion is estimated using monocular RGB camera videos, with the resulting depth recreating the entire scene, including occlusions, using the imperfect depth prediction.

Experimental design

Experiments are valid.

Validity of the findings

Results seem to be valid. The use of additional metrics, addition of confidence would improve evaluation of findings.

Additional comments

1. Improper formatting of graphs 4, 6, 7, 8; unlabeled axes; units; error bars.
2. Line 349 “Thus” → “This”
3. Dataset split is not defined, what percentage of the dataset is used as training, testing, validation.
4. Because live recording of a camera feed is used. What are real-world applications of the framework, how viable in terms of performance (FPS) is it?
5. In experimental results you specify input frames as 416x128, which corresponds to 50000 points in pointcloud. Assumptions are made that this is the reconstructed pointcloud density. However, this is the first time it is mentioned. Specify it earlier as it is important to understand this information when the method is being described.
6. Euler distance as pointcloud estimation is used. While it is valid estimation metric in the specific application case that the article uses, Chamfer Distance and Earth Movers Distance are more widely used metrics when dealing with pointclouds. Adding comparisons using those metrics would allow researches working in the field more easily evaluate your results.
7. Hard to tell what the reader is looking at in figure 5, some visual aids would be helpful to parse the visuals.
8. Substantiate your conclusion assertions using experimental results.

---

## Round 0.2 · Major Revisions

The authors should once again check the comments of reviewers and revise the paper more carefully while addressing each comment in a thorough and rigorous way. More current research should be discussed in the related works section. Before the paper contribution in Section 1, the author should clearly mention the research gap available in the literature and the importance of the proposed model. The limitations of the proposed study need to be discussed before the conclusion. The conclusions should be based on the results of this study.

Reviewer 1 ·

Basic reporting

-

Experimental design

-

Validity of the findings

-

Additional comments

Most of my remarks have not been taken very seriously and the article was changed only slightly, thus the original negative aspects still remain and I cannot recommend processing it further.

Reviewer 2 ·

Basic reporting

Article proposes a two-tier depth estimation and pointcloud reconstruction capable of rebuilding occluded data from a complex scene. Where depth, camera pose and egomotion is estimated using monocular RGB camera videos, with the resulting depth recreating the entire scene, including occlusions, using the imperfect depth prediction.

Experimental design

Experiments are valid.

Validity of the findings

Findings are valid and apt for publication.

Additional comments

Authors seem to have addressed all issues raised in previous review. No new issues have been found. Article is ready to be published.

---

## Round 0.3 · accepted · Accept

The article is accepted for publication.

[Reviewer 1 ·

Basic reporting

no comment

Experimental design

no comment

Validity of the findings

no comment

Additional comments

My review remarks have been addressed, therefor I can recommend to accept this paper.
Some English grammar and style issues remain so I would recommend getting a native speaker to proofread the paper (I've marked the review as minor - no need to send me for review)